# A Review of Intermittent Poisoning to Mitigate Toxic Plant-Induced Disease in Livestock

**DOI:** 10.3390/vetsci12010013

**Published:** 2024-12-31

**Authors:** Bryan L. Stegelmeier, T. Zane David, Kip E. Panter, Kevin D. Welch, Edward L. Knoppel

**Affiliations:** Poisonous Plant Research Laboratory, USDA Agricultural Research Service, Logan, UT 84341, USAkpanter449@gmail.com (K.E.P.);

**Keywords:** poisonous plants, intermittent poisoning, locoweed, lupine, *Astragalus*, *Oxytropis*, Lupinus, crooked-calf disease

## Abstract

Toxic plants can invade, expand through and dominate plant communities in pastures and ranges. Subsequent poisoning or the risk of poisoning often precludes using those properties. As avoiding exposure may not be possible, various strategies have been developed to minimize toxic plant consumption, toxicity and disease. Depending on the plant, the toxins and animal species, intermittent poisoning may be a useful method to avoid the costs of poisoning and use available forage. Some plant toxin-induced diseases require extended exposures to produce disease. Locoweed-induced neurologic disease and lupine-induced birth defects are two plants that may be controlled with intermittent poisoning. Select other toxic plants are also presented with discussion about their suitability or lack of methods for intermittent poisoning to be effective.

## 1. Introduction

In the 1980s the annual cost of poisonous plants to the livestock industry in 17 western United States was estimated to be 750 million USD [1]. For just half the country, that would be over 1.5 billion USD when adjusted for inflation. To avoid or minimize these losses, various treatments and management strategies have been developed. Proven strategies include several that alter plant toxicity or palatability. Some toxic plants can be reduced or eliminated by decreasing populations with herbicides. For example, poison hemlock (*Conium maculatum*) is easily controlled with most herbicides [2]. Other plants can be controlled by removing them mechanically with a shovel or cultivator. Toxic cover crops like pyrrolizidine alkaloid (PA)-containing rattlebox (*Crotalaria spectabilis*) can be controlled with cultivation, appropriate crop rotation and aggressive weed control, especially in marginal lands [3]. Toxic plants can also be excluded from fields and pastures by competitive removal with other forages. For example, toxic bindweed (*Convolvulus* spp.) is resistant to most herbicides, but it can be overgrown and excluded from fields using dense triticale seeding. Triticale, a wheat/oat hybrid, grows quickly and competes with bindweed for nutrients, ground space and sunlight [4]. Alternatively, poisoning can be controlled by altering animal factors. This might be as simple as altering grazing schedules. Some plant phenotypes are less toxic and animals can avoid exposure during periods of toxic/palatable phenotypes and use those areas later when less toxic phenotypes are available. With some toxic plants, resistant animals might safely use infested pastures or ranges. For example, sheep are relatively resistant to pyrrolizidine alkaloid poisoning and they have been used in many countries to graze *Echium* spp.-infested fields [5]. Alternatively, the toxicity of some plants can be altered by harvesting, drying or processing forages using methods that degrade or dilute the toxin. Many toxic plants such as arrowgrass (*Triglochin elata*) contain cyanogenic glycosides which are much less toxic and generally used safely when they are made into hay [6]. Of course, the most effective and often most expensive strategy is to avoid exposure altogether.

Historical experience and close monitoring of clinical poisoning combined with the understanding of plant toxins and their mechanism of action is essential in determining and optimizing exposure duration and frequency as a viable strategy to control poisoning. Not all plant toxins are cytotoxins. Some inhibit enzymes, bind or block receptors, or compete with and block nutrient use. Consequently, poisoning has to be extended until function is adequately disrupted to alter cell function and viability. This is clinically seen as poisonous plant-induced disease. In just two toxic plants identified has it been experimentally shown that their progression and clinical disease can be minimized by limiting poisoning durations separated by ample recovery periods.

## 2. Locoweeds

About 20 *Astragalus* and *Oxytropis* spp. contain mannosidase inhibiting swainsonine, an indolizidine alkaloid, which produces neurologic locoism (Figure 1). Locoism has an insidious onset that does not become apparent until susceptible animals have grazed the plant for several weeks. Initial clinical signs of anorexia, lethargy, and muscular weakness are first apparent after 12 to 14 days of exposure. An additional two weeks of continued poisoning are required to produce neurologic disease—intention tremors, proprioceptive deficits, and reluctance to move. Microscopic visceral lesions can be detected with shorter poisoning durations (10–12 days). At as soon as 10 days of continuous ingestion, there is subtle vacuolation of the urinary tract uroepithelium. Later, after about 14 days, the epithelia of pancreas, thyroid and kidneys are also vacuolated. These lesions quickly resolve (many tissues within days) after discontinuing exposure [7,8]. Four to five weeks of continued locoweed exposure produces severe neuronal vacuolation, degeneration, pyknosis and death with axonal degeneration and dystrophy developing in the cerebellum and basal ganglia. If severe these lesions are irreversible [7].

It has been hypothesized that exposing animals for brief intervals interspersed with recovery or withdrawal periods may minimize or avoid the permanent effects of locoweed poisoning. The duration and recovery times required to minimize permanent locoweed-associated lesions and disease are unknown. As reported previously, in a study to better define these intermittent exposures and recovery periods, 10 groups of lambs were dosed with *Oxytropis sericea* at 1.0 mg swainsonine/kg body weight/day for a total of 45 days [9]. The locoweed duration varied between 0, 3, 5, 9, 15 and 45 days. Each of these durations were followed by 7- or 14-day recovery periods. This design included both a positive control (45 days locoweed with no interruption) and a negative control group (lambs were gavaged with alfalfa for 45 days—0 days locoweed).

The uninterrupted locoweed (positive control) lambs were all reluctant to stand, anorexic and had prominent intention tremors when they moved [9,10]. No clinical signs were identified in the 3- and 5-day duration exposure groups for both 7- and 14-day recovery times. The lambs in the 9-day exposure group had minimal intention tremors in about half the lambs with 7-day recovery times. The clinical signs in the 15-day exposure group were severe and were similar to the positive control (45-day) lambs. All durations of locoweed lambs gained less weight than the negative control lambs. However, the lambs in all interrupted groups (both 7- and 14-day groups) gained more than the positive control. These findings indicate that clinically intermittent locoweed ingestion reduces weight gain, but less so than uninterrupted poisoning. The reduction is proportional to the dosing duration. This indicates that shorter durations had less impact on weight gains.

Histologic changes in the positive control wethers included vacuolation, eosinophilic swelling, pyknosis, and necrosis of neurons in the cerebellum, especially cerebellar Purkinje cells (Figure 2A), basal ganglia, hypothalamus, medulla and ventral horns of the spinal cord. Dystrophic axons (spheroids) were present in the white tracts of the cerebellar peduncles and white tracts of the medulla. Vacuolation in visceral tissues resolved with minimal vacuolation of the exocrine pancreas. Sheep that were dosed with durations of 9 and 15 days had less severe Purkinje cell vacuolation (Figure 2B) with minimal neuronal pyknosis and no obvious axonal dystrophy. Animals dosed at durations of 5 days or less (with both 7- and 14-day recovery periods) and the negative controls had no or minimal Purkinje cell vacuolation (Figure 2C,D). Furthermore, no neuronal pyknosis and axonal dystrophy were observed in these lambs.

Locoweed-induced permanent lesions include neuronal swelling and vacuolation followed by severe vacuolation, chromatolysis and esosinophilia, pyknosis and necrosis [10,11]. In this study, lambs dosed at durations of 5 days with 7- and 14-day recovery periods had neuronal changes that were less severe and more likely to resolve. Shorter durations did not have neuronal changes. This indicates that lambs exposed to locoweed for durations of 5 days or less with 7-day withdrawal periods are not likely to develop permanent locoweed-induced neurologic disease.

Previous dose-response studies using sheep and cattle suggest that cattle are likely to have similar responses to intermittent poisoning [8]. Studies also suggest that higher doses do not result in additional or more severe lesions [10,12]. Intermittent locoweed exposure often naturally occurs in locoweed endemic areas. These animals are often moved between different ranges, making it less likely for livestock to be exposed to locoweed for durations of longer than 45 to 60 days. Such intermittent poisoning tends to be followed by 9 or 10 months of recovery. This seasonal locoweed poisoning eventually results in permanent neuronal damage. After several seasons, such repeatedly poisoned animals have reduced numbers of cerebellar Purkinje cells (Figure 3). Microscopically, this change is best assessed in the central cerebellar lobe. Affected cattle lose condition. Though they may recover reproductive function, they will have decreased ability to move, forage and lactate. This results in small, long-haired calves that take a long time to finish. The affected cows often have behavioral changes, including becoming aggressive and difficult to handle. If intermittent poisoning is used, monitoring animals for such cumulative changes will be helpful in assessing management and optimizing frequency, duration and clearance periods.

Horses are highly susceptible to locoweed poisoning and should never be allowed to graze locoweed for any extended duration. Horses readily ingest locoweed even when other forages are available. Previously poisoned horses may recover reproductive function, but they remain neurologically compromised and should not be used for work [11,13,14].

## 3. Lupines

Lupines are common plants found in most continents in many different habitats and elevations. They are also drought-tolerant legumes that in many locations provide relatively high protein forage when other options are not available (Figure 4). However, they are toxic as they contain quinolizidine and piperidine alkaloids. More than 150 different quinolizidine and even more piperidine lupine alkaloids have been identified. These all have differing affinities to different species and acetylcholine receptor subunits and configurations, resulting in varying species-specific toxicities and clinical diseases [15].

Since the late 1800s, acute lupine poisoning has been reported in sheep, cattle, and horses. Sheep were historically the most susceptible, and in some cases hundreds of sheep have been fatally poisoned from consuming highly toxic pods and seeds. Such large poisonings generally occurred in the fall, when hungry animals were unloaded or trailed into heavy lupine patches. Acute neurologic poisoning also occurs when animals are fed lupine-contaminated hay or when snow makes other forages less available. Due to changes in grazing patterns, stocking, and transportation, such epidemic fatal poisonings are now rare. Most producers know not to expose hungry animals to dense lupines or feed lupine-contaminated hay. Cattle are less susceptible to acute poisoning and lupine-associated fatalities in cattle are infrequent. Consequently, cattle are often used to graze lupine ranges and, in some circumstances, they are unavoidably fed native lupine hay. Lupine forages are highly nutritious and they are often consumed when other forages are senescent, dry and coarse, making them less palatable with little nutritional value [16].

Though acute poisoning is infrequent in cattle, lupine ingestion produces an economically devastating syndrome of birth defects known as ‘crooked calf disease’ (CCD). Dr. Wayne Binns first described CCD and identified its association with lupine consumption in 1961 [17]. As shown in Figure 5, crooked calf disease is a collection of congenital malformations of the axial and appendicular skeletal system of newborn calves. It is caused by maternal lupine ingestion during the first trimester of pregnancy. The appendicular lesions are characterized by flexure abnormalities (contractures), often with lateral rotation and bowing of the forelimbs and swelling and fusion of the joints (arthrogryposis) (Figure 5A). The foreleg is most often severely affected, resulting in the affected calf’s inability to stand and nurse. Similar, less frequent and less severe rear limb lesions may also be present. Axial deformations include scoliosis, lordosis or kyphosis (deviations or abnormal alignment of the cervical, thoracic, or lumbar vertebrae) (Figure 5B). Other associated lesions include deformities and bending of the ribs, cranium, hooves, and sacral vertebrae (kinky tail). Cleft palate may occur with any of these other skeletal defects, depending on the gestation age associated with maternal lupine ingestion (Figure 5C) [16].

Of the hundreds of lupine alkaloids, two have been identified as the primary causes of CCD. These include anagyrine, a quinolizidine alkaloid, and ammodendrine, a piperdine alkaloid. These both have unique affinity and avid binding to fetal acetylcholine receptors. This results in inhibition of intra-uterine fetal movement with little maternal toxicity [15]. Extensive fetal ultrasonic studies have conclusively proven that alkaloid impairment of fetal movement during specific gestational periods is the cause of CCD. This was confirmed using other plant alkaloids and chemical anticholinergics that block AChR and inhibit fetal movement. Gestational timing and the duration and extent of fetal paralysis determines the types and severity of congenital defects [16]. Fetal movement must be inhibited continually for 10 to 14 days or longer during the susceptible gestation age (days 40 to 100) to result in CCD. Intermittent exposure with relatively short poisoning durations followed by lupine-free clearance periods may be a useful management tool to help minimize the effects of grazing lupine pastures and ranges as well as lupine hay in susceptible pregnant cattle.

To test the effectiveness of intermittent lupine grazing with pregnant cattle, time-bred heifers were treated with ground lupine for 30 days [18]. Briefly, 14 time-bred heifers were divided into three groups. All were treated for 30 days with ground lupine or an alfalfa control between gestation days 40 and 70. The positive control was dosed for 30 consecutive days with lupine. The intermittent group was dosed with lupine for 10 days followed by 5 recovery days two times for a total of 30 days. The negative control group was dosed with ground alfalfa for 30 consecutive days. The heifers treated with the continuous lupine dose developed severe CCD and the most severely affected calves were euthanized. Other, less severely affected calves were permanently crippled. The heifers treated with the interrupted lupine dose developed minimal or no congenital malformations consistent with CCD. These calves could stand and nurse and all recovered within several months. The heifers dosed with alfalfa as a negative control delivered normal calves. These clinical findings were supported by toxicologic and ultrasonic studies, as there was an inverse correlation between serum anagyrine (the primary teratogen in this lupine) concentrations and fetal movement. This association was especially evident in the intermittently dosed heifers, where fetal movement quickly returned to normal after lupine dosing stopped and remained normal until lupine treatment resumed. This suggests that a 10 day on and 5 day off grazing rotation could be used to reduce the incidence and severity of CCD. There is no experimental or field data yet to determine if shorter durations or longer recovery periods might completely prevent CCD.

In another study, shorter poisoning durations were tested in pregnant goats dosed with wild tobacco (*Nicotina glauca*). Wild tobacco contains teratogenic anabasine, which has been extensively used to study the mechanisms and repair of lupine-induced terata in goats [16]. In this study, pregnant goats were treated between gestation day 30 and 52 days with ground *Nicotina* for 5 days followed by 2 clearance days. Though the serum anabasine concentrations negatively correlated with fetal movement, the resulting fetuses had birth defects that had similar incidences and severities of cleft palate, appendicular angular deformities, scoliosis, lordosis and torticollis to animals dosed continuously [19].

These studies suggest intermittent lupine poisoning with at least 5 days for clearance and recovery is essential to restore fetal movement and minimize lupine-induced CCD. The most common method to avoid CCD is to keep susceptible cattle (pregnant in early gestation) away from teratogenic lupines. Although wild tobacco and poison hemlock (*Conium maculatum*) sporadically produced CCD-like axial and appendicular congenital birth defects in various species including cattle, these incidents are sporadic, and the plant populations are generally controlled with herbicides.

## 4. Other Plants That Might Be Managed Using Intermittent Exposures

### 4.1. Thiaminase-Containing Plants

Historically, thiamine deficiency has been associated with ruminant polioencephalomalacia (PEM). This was largely based on animal response to thiamine therapy, identification of rumen thiaminases, and decreased concentrations of thiamine-based cofactors. Experimental amprolium-disrupted thiamine metabolism in sheep produced PEM in about 40% of the animals; however, thiamine deficiency was not confirmed in affected animals, making its role in PEM questionable. Plants reported to contain thiaminases such as *Kochia scoparia* have been reported to cause PEM in cattle [20,21]; however, *Kochia* also commonly accumulates sulfates. Contrary to historical publications and chapters, sulfates seem to be the most likely cause of ruminant PEM. For example, blind staggers (PEM associated with selenium-accumulating plants) were initially attributed to plant thiaminases; however, PEM was never produced in plant feeding trails. Sulfate did consistently produce PEM, and water in blind stagger endemic areas often has high sulfate contamination [22]. Additionally, sulphate-induced PEM shows normal dietary and tissue thiamine concentrations [23]. High selenium or indicator plants are generally unpalatable and poisoning in those seleniferous areas is not caused by indicator plants, but by ingestion of palatable, selenium-rich plants.

Thiaminase poisoning is very different in horses and possibly other monogastric animals. Bracken fern (*Pteridium* spp.) and horsetail (*Equisetum* spp.) (see Figure 6 for both) contain thiaminases, and when horses chronically ingest these plants (whether on ranges/pasture or when they consume contaminated hay), poisoning results in thiamine (vitamin B1) deficiency and subsequent neurologic disease. Disease development requires continuous exposure and ingestion for more than a month. The subsequent neurologic disease in horses has been called bracken staggers. It is characterized by loss of condition, ataxia and stumbling. With continued exposure it can progress to weakness, inability to stand, tremors and ultimately coma, convulsions and death. Thiamine therapy is often helpful, but preventing access is the recommended management [24,25]. These neurologic lesions are permanent and previously affected animals are not sound and should not be used for work. Though no studies have been conducted on the effect of intermittent exposure, it is logical that intermittent exposure for short durations and periods for clearance and recovery might decrease both morbidity and mortality. Bracken fern contains several additional potent toxins that cause various syndromes in other species. In ruminants these include bone marrow suppression, hemorrhagic disease and uroepithelial neoplasia. Such potent toxins are best avoided by avoiding exposure.

Horsetail also contains thiaminases and poisoning has been reported in horses. It also contains oxalates, which are mostly insoluble, resulting in crystalline mucosal irritation and hyperemia. This gingivitis generally makes horsetail unpalatable. However, when other forages are not available or when it is included in hay, horses will eat it. Poisoned horses develop incoordination similar to bracken staggers. Poisoning is also chronic, requiring several months’ exposure to develop disease [26]. As with bracken fern, the effects of intermittent exposure have not been studied. However, shortened poisoning durations and recovery periods theoretically would allow thiamine concentrations to recover, reducing toxicity. More work is needed to establish both allowable durations and required recovery periods.

### 4.2. Calcinogenic Glycoside-Containing Plants

Historically, several noxious plants (Solanum glaucophyllum (Figure 7), *S. malacoxylon*, *Cestrum diurnum*, *Stenotophrum secundatum*, and *Trisetum flavenscens*) contain steroidal glycosides containing derivatives of vitamin D3. These glycosides are hydrolyzed in the gastrointestinal system and absorbed. In the intestine, they activate calcium absorption, resulting in hypercalcemia and hyperphosphatemia. The excess overwhelms tissue and cellular Ca++ accommodation and excretion, resulting in deposition in tissues (calcinosis). Primarily affecting ruminants, poisoning has also been reported in quail, chicken, rats, and rabbits. Though animals may eat standing plants in pastures and ranges, most poisoning occurs when they are harvested and consumed as contaminated hay. As mineralization begins in muscular arteries, heart, lungs and kidneys within days, if intermittent exposure is to be successful, the exposure durations would have to be short (1 or 2) days and clearance periods long enough for the hypercalcemia to resolve [27]. More work is needed to determine if minimally mineralized vessels and other tissues are reversible and to identify the optimal duration and recovery periods.

## 5. Toxic Plants That Cause Acute Damage Unlikely to Benefit with Intermittent Exposure

### 5.1. Myotoxic Trematone-Containing Plants

White snakeroot (*Ageratina altissima*—Figure 8) and rayless goldenrod (*Isocoma pluriflora*) poison most livestock species, producing skeletal and cardiac myonecrosis. Trematol, the proposed toxin, is a mixture of benzofuran ketones that requires exposure durations of 10 days or more to produce microscopic lesions producing serum biochemical indicators of increased creatinine kinase activities and troponin concentrations. With longer duration, some animals develop tolerance to additional doses. Novel exercise exacerbates toxicity compared to training, which negates previous exacerbation. This and the similarity with white muscle disease suggest that the damage is associated with oxidative damage. Repeated exposure may decrease susceptibility and reduce poisoning [28,29]. Some have suggested that doses are cumulative with interrupted exposures, even when interrupted by several days, combining to form toxic doses [30]. More work is needed to determine what doses and durations, intermittent or not, are toxic or if they result in tolerance and protect against subsequent exposures.

Other myotoxin plants such as *Cassia occidentalis* and *Thermopsis montanae* are also highly toxic, as animals develop disease within hours of ingestion and disease is generally quickly fatal [31,32]. This suggests that intermittent poisoning would probably not be helpful in their management.

### 5.2. Neurotoxic Plants

There are a number of highly toxic, neurotoxic plants to which any exposure should be considered dangerous. Some of these are pretty good forages and consequently they are often eaten. For example, tall larkspur (*Delphinium barbeyi*, *D. occindentale*, *D. glaucum*) is palatable and often eaten by cattle (Figure 9). This consumption is often cyclic as the positive post-ingestion consequences (common in high protein forbs) promote further ingestion until animals obtain a toxic dose, resulting in negative post-ingestive consequences. These cause cattle to avoid eating larkspur [33]. This cyclic poisoning is generally not fatal. Fatal poisoning occurs when cattle eat too much too quickly. Consequently, most poisonings occur when hungry, naïve animals are moved into larkspur pastures or ranges. Alternatively, some other event, possibly a storm, keeps animals away from alternative forages or some unidentified plant or animal factor increases larkspur palatability. Historically different mineral supplements have been suggested to alter larkspur toxicity, but field studies are needed to document protection [34]. Intermittent exposure would repeatedly put susceptible cattle in larkspur-contaminated ranges/pastures where they might eat toxic doses, making that a risky option.

Similarly, highly toxic death camas (*Toxicoscordion* spp.), water hemlock (*Cicuta douglasii*), various cardiac glycoside-containing plants, and cyanogenic glycoside-containing plants are highly toxic and even a single dose has the potential to be fatal. Consequently, poisoned animals are not likely to benefit from intermittent exposure.

### 5.3. Hepatotoxic Plants

Cocklebur (*Xanthium* spp.), lanceleaf sage (*Salvia reflexa*—Figure 10), lantana (*Lantana camara*), alsike clover, and pyrrolizidine alkaloid-containing plants have a variety of toxins, but most are potent alkylating toxins that damage hepatocytes, resulting in hepatic necrosis, cirrhosis and loss of function. It has been suggested that using less susceptible species at low doses may be a more likely strategy. For example, sheep and goats are both more resistant to pyrrolizidine poisoning and short-term exposures may allow animals to recover with minimal biochemical and functional damage [35]. For several of these, damage often progresses after exposure is discontinued. Consequently, intermittent poisoning is not likely practicable.

## 6. Conclusions

In some situations, intermittent grazing has been experimentally shown to reduce the adverse effects of locoweed and lupines. Additional work is needed to better define exposure limits, duration limits and recovery times for thiaminase-containing plants, plants that alter calcium metabolism and other endocrine and nutritional disrupters. The practical implementation of such intermittent or limited exposures is costly and labor-intensive. Moving animals every couple of days is often not possible. Additionally, in many locations, poisonous plant infestations are widespread and finding and maintaining “safe” pastures or ranges is expensive or even impossible. The result is that intermittent or controlled toxic plant exposures are rarely implemented. Certainly, limited use such as combining short exposures with other measures such as using resistant species, limited herbicide control, and liberal use of supplemental forages or neutralizing compounds would be more easily implemented. Successful strategies to minimize poisoning will require implementation of all available tools. Intermittent exposures and limited or short-duration poisoning is yet another option. Each problematic plant will need to be evaluated according to its risk and the negative effect on production. Of course, avoiding exposure will always be the recommended practice for highly toxic plants.

## Figures and Tables

**Figure 1 vetsci-12-00013-f001:**
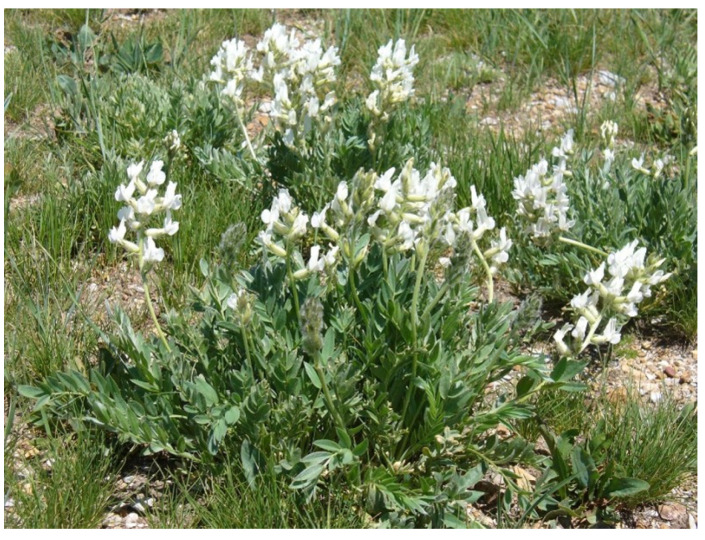
*Oxytropis sericea* is a deep-rooted legume with immense, long-lived seed banks that allows constant populations that poison livestock and wildlife. The locoweeds are native species that are commonly found in western North America. They are drought-tolerant, deep-rooted perennials whose extensive, long-lived soil seed banks make herbicide control difficult and temporary. Similar toxic plants have also been identified in South America and Asia.

**Figure 2 vetsci-12-00013-f002:**
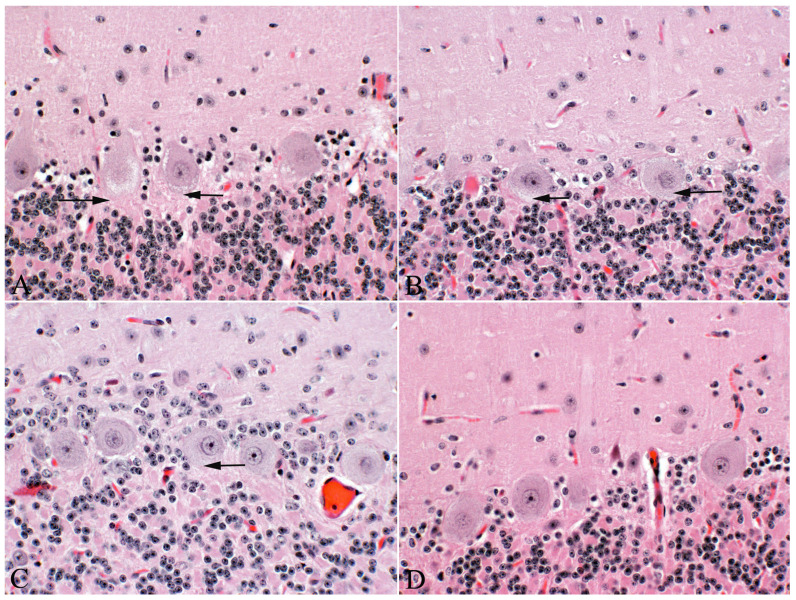
Photomicrograph of central lobe of the cerebellum of locoweed-poisoned sheep with varying dosing durations with a total of 45 total dosing days. (**A**)—lamb dosed continuously with locoweed for 45 days with no recovery periods. Notice the marked, fine, granular vacuolation in Purkinje cells (arrows). Small numbers of pyknotic Purkinje cells were observed. (**B**)—lamb similarly dosed for 45 days with durations of 15 days (3 courses) each followed by a 14-day recovery period. Notice the Purkinje cells have similar but less severe fine granular vacuolation (arrows). No pyknotic neurons were identified. (**C**)—Lamb similarly dosed with locoweed for 45 days with durations of 5 days (9 courses) each followed by a 14-day recovery period. Notice the minimal but real fine granular cytoplasmic vacuolation (arrow). No pyknotic neurons were identified. (**D**)—Negative control lambs that were similarly dosed via oral gavage with ground alfalfa for 45 days. Notice the absence of Purkinje cell change. Similarly, no pyknotic neurons were identified.

**Figure 3 vetsci-12-00013-f003:**
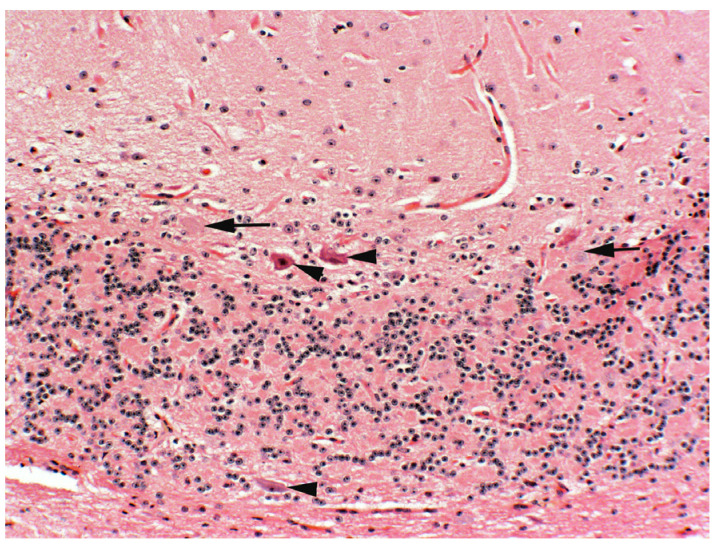
Photomicrograph of cerebellar Purkinje cells from a cow that was previously poisoned by Green river milkvetch (*Astragalus pubentissimus*) for at least 5 years. Before necropsy, she was thin, wasted, and anxious and aggressive to handlers. Her 6-month-old calf was small (about 150 kg or 30% smaller than her herdmates) with long hair. Notice the paucity of cerebellar Purkinje cells in the central cerebellar lobe (arrows indicate empty baskets in locations of missing Purkinje cells). The arrow heads are degenerative pyknotic Purkinje cells. The cerebellar crus and white tracts in the medulla contained numerous axonal spheroids (not shown).

**Figure 4 vetsci-12-00013-f004:**
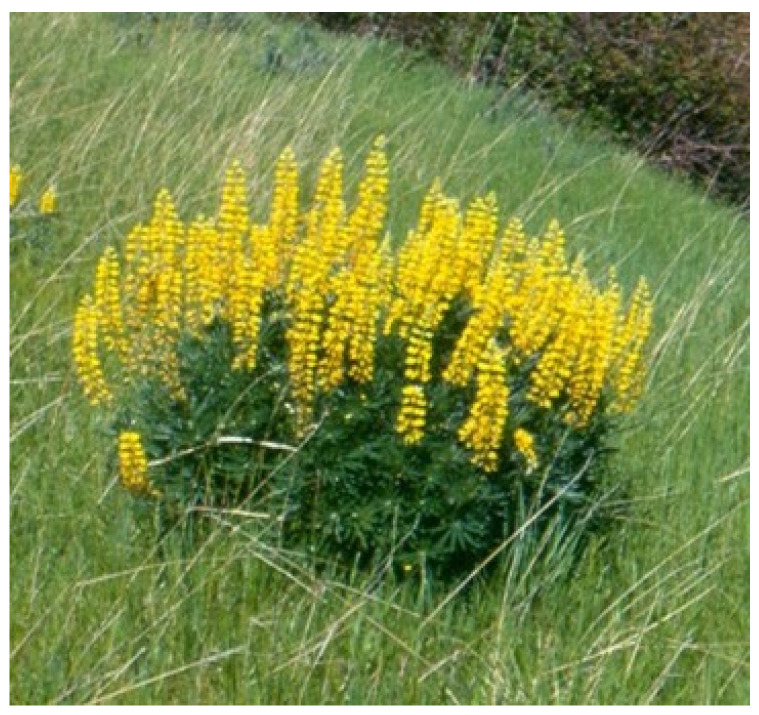
*Lupinus sulphureus* is a deep-rooted perennial that often dominates arid range plant populations. It has seven distinct chemical profiles or chemotypes that range from non-toxic to extremely teratogenic. This teratogenic chemotype produces a high incidence of congenital appendicular and axial skeletal deformities. Morbidity can be high, almost epidemic, affecting nearly all calves in exposed herds. Chemical analysis is essential to determine the plant chemotype and the subsequent risk of poisoning. Teratogenic lupines have been identified in many parts of western North America.

**Figure 5 vetsci-12-00013-f005:**
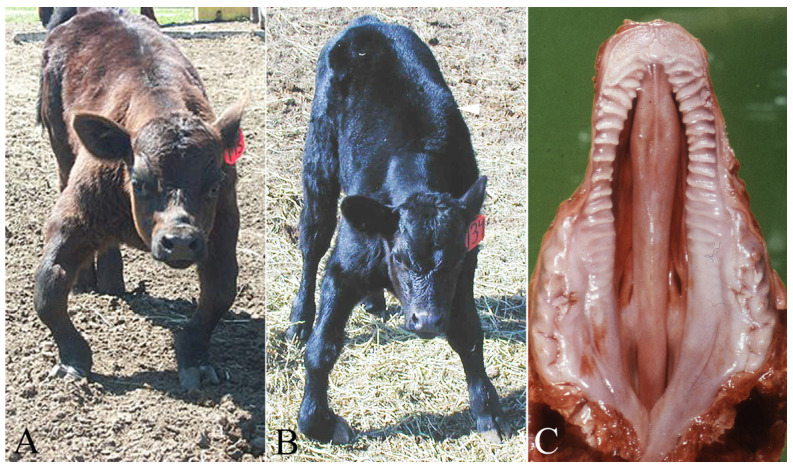
Lupine-induced congenital birth defects. These are all clinical cases where the pregnant cows were exposed to various lupines during gestation days 40 to 100. (**A**)—“Crooked calf” that has marked contraction and alkylosis of the front legs. There is also valgus rotation of both legs. (**B**)—Less severe “crooked calf” that has contraction and alkylosis of the front right pastern with minimal valgus rotation of both legs. The arrowhead is an axial lesion as the thoracic vertebrae have a left lateral deviation (scoliosis). (**C**)—Complete cleft palate involving both the soft and bony palate. Such lesions are often associated with gestational lupine ingestion between 40 and 50 days.

**Figure 6 vetsci-12-00013-f006:**
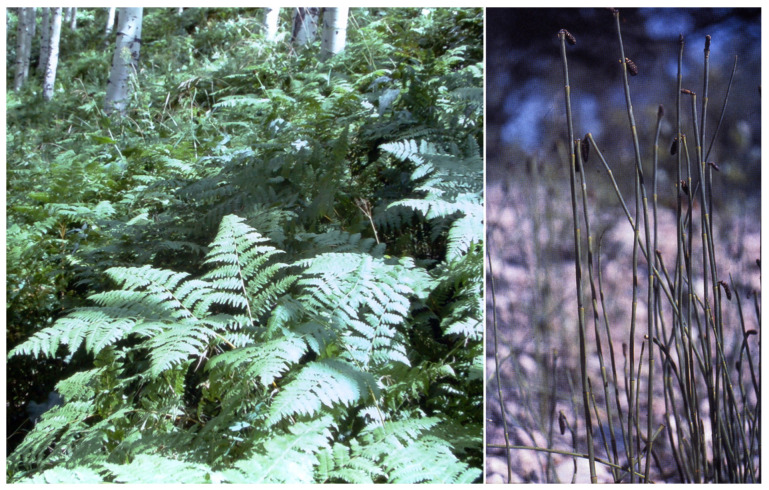
Bracken fern (*Pteridium aquilinum*—left) and field horsetail (*Equisetum arvense*—right) both contain thiaminases that produce neurologic disease in horses. Though it has been suggested that they might cause polioencephalomalacia in ruminants, that has not been experimentally replicated. Bracken fern also contains carcinogens (several ptaquilosides and braxins). They also contain various other toxins (cyanogenic glycosides, flavonoids (quercetin and rutin), tannins, and p-hydroxystyrene glycosides (ptelatoside-A and ptelatoside-B). The effects of these toxins seem to be cumulative, making intermittent exposure risky.

**Figure 7 vetsci-12-00013-f007:**
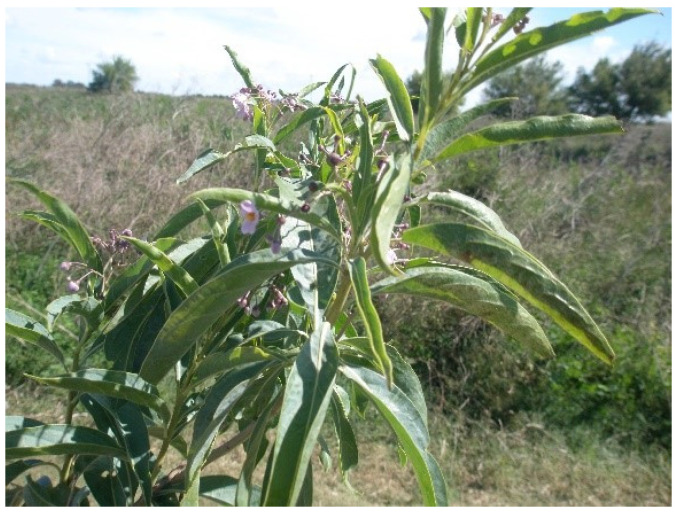
Waxyleaf nightshade (*Solanum glaucophyllum*) is a South America native that, like *S. malacoxylon*, *Cestrum diurnum*, *Stenotophrum secundatum*, and *Trisetum flavenscens*, contains 1,25 dihydroxycholecalciferol glycosides that when hydrolyzed form active vitamin D3. When livestock, mostly cattle, ingest the plant, poisoning results in extensive soft tissue mineralization with subsequent vascular and muscular calcification.

**Figure 8 vetsci-12-00013-f008:**
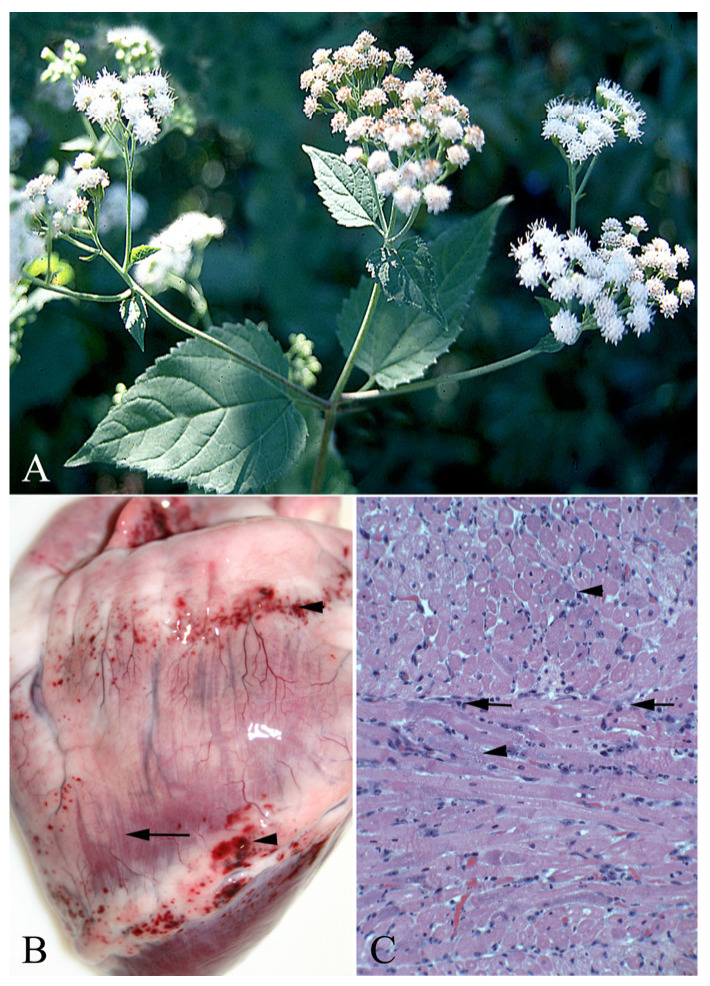
(**A**)—White snakeroot (*Ageratina altissima*) is a North American plant that, though it has been used medicinally, poisons livestock and humans. It is a perennial that spreads via seed and fibrous rhizomes. Historically, it causes milk sickness poisoning as it contaminates milk, poisoning nursing neonates or humans. (**B**)—Heart of a horse poisoned with white snakeroot. Notice the marked vascular congestion with hemorrhages along the coronary groove and atrial-ventricular margins. The epicardial lymph vessels are enlarged. (**C**)—Myocardium of the same horse in (**B**) that has myocardial degeneration and necrosis with loss of striation, clumping of myocardial proteins (arrowheads) and inflammatory cell infiltrates (arrows). These lesions developed within several days of exposure, suggesting that such damage can occur with relatively short exposures. As such lesions result in fibrosis and decreased function, such poisoning probably results in permanent functional impairment.

**Figure 9 vetsci-12-00013-f009:**
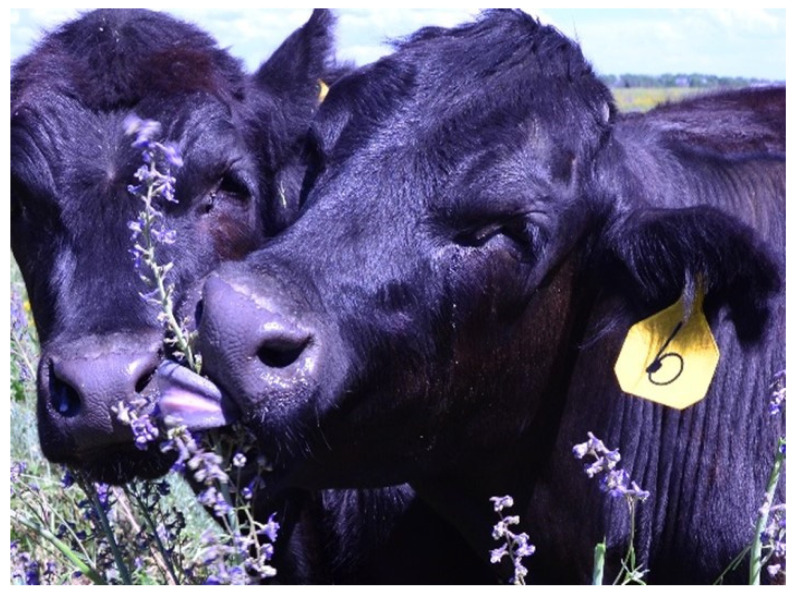
Cattle eating tall larkspur (*Delphinium barbeyi*), which like many other larkspurs contains several different diterpenoid alkaloids that bind and block nicotinic acetylcholinergic receptors. The effect is weakness and paresis leading to recumbence, rumen tympany, respiratory failure and death. Cattle are uniquely susceptible, and they are fatally poisoned when they eat too much too quickly.

**Figure 10 vetsci-12-00013-f010:**
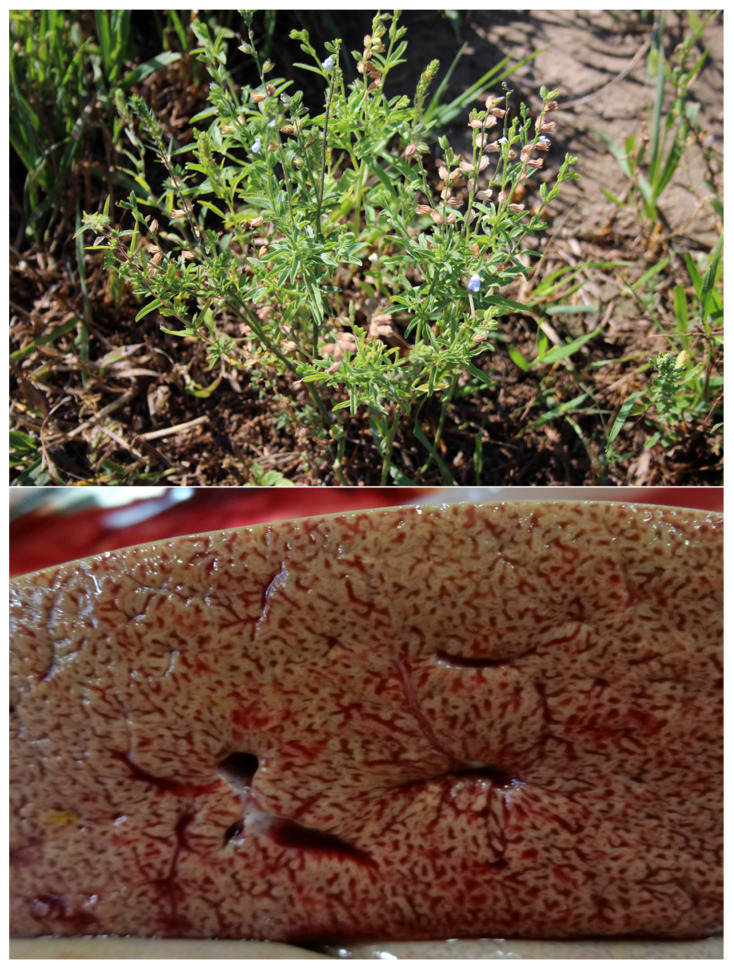
Lanceleaf sage (*Salvia reflexa*), one of the highly potent hepatotoxic plants that cause acute liver necrosis. Others are cocklebur (*Xanthium* spp.), lantana (*Lantana camara*), alsike clover, and pyrrolizidine alkaloid-containing plants. As these cause severe necrosis or continue to cause damage after exposure has ceased, intermittent exposures are probably not indicated.

## Data Availability

This data is largely published in the indicated references.

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
