# Peer review of "A Review of Intermittent Poisoning to Mitigate Toxic Plant-Induced Disease in Livestock"

_vetsci, 2024, doi:10.3390/vetsci12010013_

Round 1

Reviewer 1 Report

Comments and Suggestions for Authors

The review article “A review of intermittent poisoning to mitigate toxic plant induced disease in livestock” lists several toxic plants for livestock and wildlife animals. This is a current and interesting topic considering all the risks and consequences that can derive from animals’ poisoning after toxic plant ingestions. Authors discuss how intermittent exposure to these toxic plants can be effective in reducing their poisoning potential and effects. The work is in general well written and discussed, also if I think that, being a review, it would be interesting to look at little deeper into the toxicity-mechanism of such plants. Moreover, some details about the geographic distribution of the plants taken into account should be added.

Moreover:

Line 83-121: please, add bibliography indicating where these experiments are reported.

Lines 408-410: please rewrite the sentence explaining better the conclusions.

For the reasons written above, I believe that the manuscript can be accepted for its publication in “Veterinary Sciences”, after the minor revisions described.

Author Response

Reviewer 1:

Comment 1: “it would be interesting to look at little deeper into the toxicity-mechanism of such plants. Moreover, some details about the geographic distribution of the plants taken into account should be added.”

The discussion of locoweed and lupine were expanded to better describe the mechanism of toxicity and geographic distribution of these plants.

Comment 2: Line 83-121: please, add bibliography indicating where these experiments are reported.

These references were included in the references. A footnote was included to the references to the two major locoweed and lupine studies.

Comment 3: Lines 408-410: please rewrite the sentence explaining better the conclusions

The conclusions were rewritten.

Reviewer 2 Report

Comments and Suggestions for Authors

The article tackles a real and significant issue: the impact of toxic plants on livestock and the need for effective management strategies. The economic cost cited at the beginning underscores the importance of finding solutions to this problem. The concept of "intermittent poisoning" is intriguing and relatively unexplored, making the article highly valuable to both the scientific community and livestock producers. Furthermore, the article presents a robust amount of experimental studies and evidence supporting the feasibility of intermittent management for specific toxic plants, such as locoweed and lupine. Proposing strategies that allow the sustainable use of infested pastures without causing permanent harm to livestock aligns perfectly with the current demand for sustainable agricultural practices. Once again, congratulations to the authors.

Please:

  • Standardize the development of the various topics to ensure all sections are equally detailed and cohesive.
  • Discuss additional aspects related to the use of pesticides/herbicides, including the risks they pose to other species, and compare these risks more thoroughly with the strategies proposed in the article.
  • Highlight the limitations of the studies conducted and suggest directions for future research, particularly experimental studies.
  • Consider introducing a table summarizing the various toxins and their main effects. This could enhance the readability and help the reader follow the text more effectively.
  • Slightly adjust the focus to emphasize the direct impact of these strategies on the daily work of rural producers or environmental managers.

These additions could make the article even more comprehensive and impactful.

Author Response

Reviewer 2:

Comment 1: Standardize the development of the various topics to ensure all sections are equally detailed and cohesive.

Two plants, locoweeds and lupines are unique as intermittent poisoning has been studied for these plants. Their discussion consequently is more extensive. The other plants are potential problems that do to their mechanism of toxicity might be allow such intermittent exposures. The text was altered to better segregate these differences.

Comment 2: Discuss additional aspects related to the use of pesticides/herbicides, including the risks they pose to other species, and compare these risks more thoroughly with the strategies proposed in the article.

As suggested in the introduction, some plants are easily controlled with herbicides. Not all such plants were included and in out opinion is not in the scope of this brief review. A reference was included as a suggestion where that information is available.

Comment 3: Highlight the limitations of the studies conducted and suggest directions for future research, particularly experimental studies.

The discussion of both the locoweed and lupine work was expanded to better highlight the limitations of those studies and what future work might better define “safe” exposures.

Comment 4: Consider introducing a table summarizing the various toxins and their main effects. This could enhance the readability and help the reader follow the text more effectively.

This was considered but seemed redundant and unnecessary is this relatively small and focused review.

Comment 5: Slightly adjust the focus to emphasize the direct impact of these strategies on the daily work of rural producers or environmental managers.

The introduction was expanded with a discussion of the costs (both direct and indirect) and difficulty of intermittent poisoning.

Reviewer 3 Report

Comments and Suggestions for Authors

The article belongs to the scope of the journal, is adequately written and contributes to knowledge in the field. Thus, it has merit to be published. 

Here are some suggestions for authors:

Regarding Solanum glaucophyllum poisoning, there are reports of lesions occurring without the manifestation of clinical signs depending on the quantity consumed. These injuries would not be reversible and therefore, after stopping consumption, when consume again, they could evolve from the initial point. This aspect could be better discussed.

In poisoning by plants of the Crotalaria genus, the occurrence of adaptation to the consumption of non-lethal doses and also the cumulative effect of the active ingredient are described. These aspects could also be considered, since intermittent consumption may be related to both conditions.

Author Response

Reviewer 3:

Comment 1: Regarding Solanum glaucophyllum poisoning, there are reports of lesions occurring without the manifestation of clinical signs depending on the quantity consumed. These injuries would not be reversible and therefore, after stopping consumption, when consume again, they could evolve from the initial point. This aspect could be better discussed.

The discussion was expanded to indicate that irreversible lesions are not likely to be candidates for intermittent exposures. With mineral accumlations, exposure and withdrawl durations are critical and there is little information on what lesions are truly reversible.

Comment 2: In poisoning by plants of the Crotalaria genus, the occurrence of adaptation to the consumption of non-lethal doses and also the cumulative effect of the active ingredient are described. These aspects could also be considered, since intermittent consumption may be related to both conditions.

The discussion on the pyrrolizidine alkaloid section was expanded. As with many toxic plants, the dose is critical, and some low doses are easily tolerated. However, the progressive nature of poisoning suggests that identifying recovery or clearance durations that would ensure safety is difficult.